# Artificial Gametogenesis and In Vitro Spermatogenesis: Emerging Strategies for the Treatment of Male Infertility

**DOI:** 10.3390/ijms26157383

**Published:** 2025-07-30

**Authors:** Aris Kaltsas, Maria-Anna Kyrgiafini, Eleftheria Markou, Andreas Koumenis, Zissis Mamuris, Fotios Dimitriadis, Athanasios Zachariou, Michael Chrisofos, Nikolaos Sofikitis

**Affiliations:** 1Third Department of Urology, Attikon University Hospital, School of Medicine, National and Kapodistrian University of Athens, 12462 Athens, Greece; ares-kaltsas@hotmail.com (A.K.); a_koumeni@icloud.com (A.K.); mchrysof@med.uoa.gr (M.C.); 2Laboratory of Genetics, Comparative and Evolutionary Biology, Department of Biochemistry and Biotechnology, University of Thessaly, Viopolis, Mezourlo, 41500 Larissa, Greecezmamur@uth.gr (Z.M.); 3Department of Microbiology, University Hospital of Ioannina, 45500 Ioannina, Greece; eleftheria.markou4@gmail.com; 4Department of Urology, Faculty of Medicine, School of Health Sciences, Aristotle University of Thessaloniki, 54124 Thessaloniki, Greece; helabio@yahoo.gr; 5Laboratory of Spermatology, Department of Urology, Faculty of Medicine, School of Health Sciences, University of Ioannina, 45110 Ioannina, Greece; azachariou@uoi.gr

**Keywords:** in vitro spermatogenesis, in vitro gametogenesis, male infertility, pluripotent stem cells, organoids, ethical considerations

## Abstract

Male-factor infertility accounts for approxiamately half of all infertility cases globally, yet therapeutic options remain limited for individuals with no retrievable spermatozoa, such as those with non-obstructive azoospermia (NOA). In recent years, artificial gametogenesis has emerged as a promising avenue for fertility restoration, driven by advances in two complementary strategies: organotypic in vitro spermatogenesis (IVS), which aims to complete spermatogenesis ex vivo using native testicular tissue, and in vitro gametogenesis (IVG), which seeks to generate male gametes de novo from pluripotent or reprogrammed somatic stem cells. To evaluate the current landscape and future potential of these approaches, a narrative, semi-systematic literature search was conducted in PubMed and Scopus for the period January 2010 to February 2025. Additionally, landmark studies published prior to 2010 that contributed foundational knowledge in spermatogenesis and testicular tissue modeling were reviewed to provide historical context. This narrative review synthesizes multidisciplinary evidence from cell biology, tissue engineering, and translational medicine to benchmark IVS and IVG technologies against species-specific developmental milestones, ranging from rodent models to non-human primates and emerging human systems. Key challenges—such as the reconstitution of the blood–testis barrier, stage-specific endocrine signaling, and epigenetic reprogramming—are discussed alongside critical performance metrics of various platforms, including air–liquid interface slice cultures, three-dimensional organoids, microfluidic “testis-on-chip” devices, and stem cell-derived gametogenic protocols. Particular attention is given to clinical applicability in contexts such as NOA, oncofertility preservation in prepubertal patients, genetic syndromes, and reprocutive scenarios involving same-sex or unpartnered individuals. Safety, regulatory, and ethical considerations are critically appraised, and a translational framework is outlined that emphasizes biomimetic scaffold design, multi-omics-guided media optimization, and rigorous genomic and epigenomic quality control. While the generation of functionally mature sperm in vitro remains unachieved, converging progress in animal models and early human systems suggests that clinically revelant IVS and IVG applications are approaching feasibility, offering a paradigm shift in reproductive medicine.

## 1. Introduction

Infertility remains a pressing public health challenge, affecting an estimated 17.5% of couples at some point in their lives and 12.6% in any given year [1]. Approximately half of all cases involve a male-factor component, with 10–15% of the affected men presenting with azoospermia—the complete absence of spermatozoa in the ejaculate—primarily due to failed spermatogenesis [2,3]. For this subgroup, contemporary assisted reproduction techniques, such as in vitro fertilization (IVF) or intracytoplasmic sperm injection (ICSI), offer limited options when testicular sperm retrieval is impossible, leaving donor gametes or adoption as the only alternatives [4]. The clinical and psychological burden of these circumstances has sparked an intense search for therapeutic avenues able to generate functional, patient-specific gametes.

Over the past decade, two converging research streams have shifted the ambition of artificial gametogenesis from speculation toward plausibility. First, organotypic in vitro spermatogenesis (IVS) aims to stimulate residual testicular tissue to complete the entire spermatogenic program outside the body. Second, de novo gametogenesis from pluripotent or reprogrammed somatic stem cells (collectively termed in vitro gametogenesis, IVG) aims to create sperm or even oocytes from the ground up [5]. Landmark murine studies have already produced fertile offspring from IVS- or IVG-derived gametes [6], while incremental successes have been reported in rats and non-human primates. Success in this field could provide a transformative solution for men with azoospermia, cancer survivors who lost fertility before puberty, and individuals with specific genetic disorders. Moreover, artificial gametogenesis presents a unique opportunity to enhance scientific understanding of human spermatogenesis, clarify the mechanisms underlying male infertility, and potentially enable personalized reproductive therapies. However, translating these methods to humans remains contentious: encouraging case reports of round-spermatid formation compete with equally robust data showing meiotic arrest, chromosomal errors, and epigenetic instability [7,8]. In this review, the term epigenetic denotes heritable modifications in gene expression—such as DNA methylation and histone alterations—that occur during gametogenesis without changing the underlying DNA sequence. The natural course of epigenetic reprogramming in germ cell development, including genome-wide demethylation in primordial germ cells followed by sex-specific remethylation, represents a critical biological process that must be accurately recapitulated in any artificial gametogenesis strategy. These endogenous reprogramming events are conceptually distinct from exogenous in vitro manipulations—such as the application of small-molecule epigenetic modulators—designed to emulate or induce specific epigenetic modifications under culture conditions. The ethical debate intensifies further when IVG is considered as a means for same-sex reproduction or post-mortem parenthood [9].

This review synthesizes current evidence across biology, bioengineering, and translational medicine to clarify the status of IVS and IVG as emerging solutions for severe male-factor infertility. Key technical platforms, including organotypic slice culture, three-dimensional organoids, microfluidic “testis-on-chip” devices, and stem-cell-based reprogramming, are compared, and their performance is benchmarked against species-specific milestones. Controversial findings are critically examined, particularly those related to meiotic competence and epigenetic integrity. Potential clinical applications, such as treatments for non-obstructive azoospermia (NOA), oncofertility preservation, and disorders of sex development, are mapped alongside the ethical, regulatory, and societal challenges that shape their future use. The current body of evidence suggests that while the reliable in vitro production of haploid germ cells is now achievable, the generation of fully mature, functionally validated sperm remains an unmet goal. The development of multidimensional culture systems and precise temporal control of differentiation signals are identified as key priorities for future research. By delineating both achievements and bottlenecks, this review provides a concise roadmap for the safe, effective, and ethically compliant progression of laboratory-derived gametogenesis toward clinical fertility care.

## 2. Biological Basis of Spermatogenesis and the In Vitro Challenge

Human spermatogenesis exemplifies a highly coordinated, tissue-specific stem cell system in which a rare pool of spermatogonial stem cells (SSCs) sustains the lifelong production of terminally differentiated gametes within the seminiferous tubules [10]. This process relies on seamless molecular crosstalk between germ cells and the surrounding somatic niche, including Sertoli cells, Leydig cells, peritubular myoid cells, immune cells, and components of the vascular network, all organized within an immune-privileged environment. Successfully replicating this multicellular dialog outside the body remains one of the greatest hurdles in reproductive biology [11].

### 2.1. Spermatogenesis—Cellular Choreography in the Seminiferous Tubule

Each spermatogenic wave lasts ~74 days in adult men and is initiated every 16 days, ensuring continuous sperm output [12]. The process follows a well-defined sequence of stages: spermatogonia (mitotic proliferation of spermatogonial stem cells, SSCs), primary spermatocytes (meiotic division I), secondary spermatocytes (meiotic division II), and spermatids (post-meiotic remodeling). SSCs reside on the basement membrane of the seminiferous epithelium and include several states from quiescence to differentiation commitment [13,14]. After the final mitotic division, type B spermatogonia give rise to primary spermatocytes, which traverse the tight junctions of the blood–testis barrier (BTB) to enter meiosis. Two rounds of meiosis yield haploid round spermatids. Spermiogenesis then transforms round spermatids into mature spermatozoa via acrosome formation, histone-to-protamine chromatin remodeling, flagellum assembly, and cytoplasmic shedding, culminating in the release of mature spermatozoa into the tubule lumen [15,16,17].

Crucially, germ cell development is not autonomous; it relies on intimate support from the surrounding somatic niche, especially the Sertoli cells. Sertoli cells, often termed “nurse cells,” span the seminiferous epithelium and form the BTB via specialized junctions, creating distinct basal and adluminal compartments [18]. The anatomical positioning and diverse functions of Sertoli cells are illustrated in Figure 1, highlighting their role in establishing immune privilege and shaping a specialized microenvironment for germ cell development.

Sertoli cells actively nurture developing germ cells by supplying metabolic and regulatory factors: for instance, they metabolize glucose into lactate, an essential energy substrate for spermatocytes and spermatids, and secrete transferrin to deliver iron to germ cells [18]. They also produce key growth factors and cytokines—notably stem cell factor (SCF) and glial cell line-derived neurotrophic factor (GDNF)—which promote SSC maintenance and differentiation. Additionally, Sertoli cells phagocytose residual cytoplasm from maturing spermatids, recycling nutrients and maintaining tubule homeostasis [19]. Endocrine regulation of spermatogenesis is largely mediated through Sertoli cells: follicle-stimulating hormone (FSH) acts on Sertoli cell receptors to stimulate the production of androgen-binding protein (ABP) and inhibin B, thereby concentrating testosterone within the tubule and providing feedback to the pituitary, while intratesticular testosterone from Leydig cells diffuses across the BTB and binds to Sertoli cell androgen receptors to “license” meiosis and late spermiogenesis [20].

Other somatic cell types furnish complementary support. Leydig cells in the interstitium produce pulsatile testosterone, which, as noted, is indispensable for meiotic progression and spermatid maturation via Sertoli cell signaling [21]. Peritubular myoid cells form the outer layer of the tubules, providing structural integrity and contractility to aid sperm transport; they also secrete extracellular matrix components crucial for the basement membrane and overall tubule architecture [22]. Testicular macrophages and endothelial cells further contribute by maintaining the immune-privileged status of the testis and regulating local blood flow and nutrient delivery [23]. The precise three-dimensional arrangement of these cells and germ cells within each seminiferous tububle creates localized gradients of nutrients, hormones, paracrine signals that orchestrate synchronous waves of spermatogenesis [24].

### 2.2. Bottlenecks in the In Vitro Recreation of Spermatogenesis

Replicating spermatogenesis in vitro remains one of the most formidable challenges in reproductive biology. A fundamental bottleneck lies in re-establishing a fully functional BTB. The BTB not only shields post-meiotic germ cells from systemic immune surveillance but also generates the specialized adluminal microenvironment required for meiosis. However, in organ culture systems, Sertoli-cell tight junctions often assemble incompletely or misalign, producing a leaky interface that arrests meiotic progression [25,26].

An equally pressing obstacle is the maintenance of long-term tissue viability. Whereas human germ cells require more than two months to attain complete maturation, most current culture systems preserve intact testicular tissue for no more than four to six weeks [27]. Prolonged static culture conditions lead to hypoxia, nutrient depletion, accumulation of reactive oxygen species, and dedifferentiation of somatic cells, all of which jeopardize the survival and function of SSCs.

Successful spermatogenesis in vitro also depends on the precise synchronization of endocrine and paracrine cues. In vivo, oscillatory patterns of testosterone, FSH, and retinoic acid (RA) coincide with specific stages of germ cell development, providing stage-specific signals that regulate the progression of spermatogenesis. Recreating these temporally precise pulses in vitro is highly challenging [28]. Continuous exposure to these factors often leads to premature meiotic entry or temporal desynchronization, a phenomenon exemplified by the appearance of round spermatid-like cells after merely 10–30 days in culture—far earlier than the expected 60–70 days required for complete spermatogenesis in vivo [29].

Moreover, complete epigenetic reprogramming must be supported. Accurate imprint erasure, de novo DNA methylation, and chromatin condensation depend on a finely tuned somatic–germ cell dialog and still-elusive metabolic signals [30]. Failure to replicate this delicate interplay in vitro risks producing germ cells with aberrant epigenetic marks, compromised chromatin condensation, and impaired developmental potential. These disruptions are strongly influenced by oxidative-stress-related pathways, which can impair DNA methylation and histone modification processes, ultimately compromising sperm quality and embryonic development. Kaltsas et al. recently highlighted the role of oxidative-stress-mediated epigenetic dysregulation in male infertility and transgenerational effects on offspring health [31]. This has been observed in pluripotent-stem-cell-derived spermatid-like cells, which frequently exhibit aberrant imprinting patterns, defective protamine exchange, and compromised chromatin integrity [32,33].

Finally, engineering a biomimetic three-dimensional scaffold that permits efficient gas and nutrient exchange is critical for sustained IVS. The seminiferous tubule is a convoluted cylinder roughly 150 µm in diameter, ensheathed by contractile myoid cells and closely associated with a capillary network that ensures adequate perfusion [34]. Two-dimensional monolayer cultures cannot recapitulate this architecture, and even free-floating organoids frequently become centrally hypoxic, leading to germ cell arrest at the pachytene stage of meiosis [35].

Together, these biological and bioengineering challenges illustrate the complexity of replicating spermatogenesis in vitro. Overcoming these obstacles requires integrated solutions that combine advanced scaffold design, controlled molecular signaling, and long-term culture strategies that can sustain both germ and somatic cell function.

## 3. Technical Methodologies for In Vitro Gametogenesis

Over the past decade, it has become increasingly clear that no single culture system is yet capable of reproducing the entire 64-day human spermatogenic cycle with clinically relevant efficiency. Instead, several complementary technologies—including organotypic slice culture, bioengineered three-dimensional (3D) platforms, dynamic microfluidics, somatic cell co-culture, and stage-defined growth-factor supplementation—have each succeeded in recapitulating specific facets of the testicular niche while struggling with others. The sections below critically appraise these methodologies, outline their most recent refinements, and identify key bottlenecks that must be addressed before clinical translation.

### 3.1. Organotypic Testis Culture

Organotypic testis culture represents the most direct attempt to replicate spermatogenesis in vitro by preserving the native tissue architecture. In this approach, thin slices or small fragments of testicular tissue from a donor are cultured at a air–liquid interface (ALI) or embedded in soft gels, aiming to maintain the spatial arrangement of germ and somatic support cells as it exists in vivo. The ALI method, wherein tissue slices sit at the interface of culture medium and gas, has become a reference technique for sustaining seminiferous tubule cytoarchitecture ex vivo. In rodents (especially mice), ALI culture can support the entire spermatogenic program to completion, yielding fertilization-competent spermatozoa [36]. However, translating these successes to larger mammals and humans has proven challenging.

Long-term ALI culture of prepubertal human tissue (up to 139 days) can preserve somatic cell types, including Sertoli (SOX9^+^), Leydig (STAR^+^, CYP17A1^+^), and peritubular-myoid cells. Yet germ cells fare less well: SSC populations tent to dwindle (often >50% loss), and in many cases germ cell development arrests at the spermatocyte or round spermatid stage without producing mature sperm [25,26,37]. Recent refinements have sought to overcome these limitations. Providing better oxygenation and nutrient diffusion through supportive matrices has shown promise. For example, embedding tissue slices in low-stiffness hydrogels (agarose, alginate) or using chitosan-based bioreactors improves oxygen delivery and reduces hypoxia-induced cell death. Notably, one study reported that human testis fragments from adult transgender petients cultured in a soft hydrogel produced morphologically identifiable spermatozoa after 55 days—the first claim of complete human spermiogenesis ex vivo—although the efficiency was low and independent confirmation is needed [38]. Likewise, fetal testicular tissue (~12–19 weeks gestation) appears highly permissive. Yuan et al. generated fertilization-competent round spermatids within 50 days, documented normal imprinting through single-cell RNA sequencing, and achieved blastocyst formation after round-spermatid injection (ROSI) [39]. However, the yield of haploid cells was very low (0.07–9.8%) and importantly, ethical considerations rule out the clinical use of fetal tissues.

In summary, several biological and technical limitations currently constrain the efficacy of organotypic culture IVS. Static slice cultures suffer from diffusion gradients—oxygen and nutrients become depleted and waste metabolites accumulate in the core of the tissue—leading to hypoxia, apoptosis, and compromised somatic cell function over time. The reassembly of a fully functional BTB in vitro is often incomplete or mislocalized, which blocks meiotic progression past the mid-meiosis stage. Additionally, human tissue seems intrinsically more prone to in vitro spermatogenic arrest than rodent tissue, possibly due to its longer spermatogenic cycle and greater sensitivity to suboptimal culture conditions. Nevertheless, organotypic culture remains a valuable platform for studying human spermatogenesis and testing interventions to improve it. Ongoing optimizations—including biomaterial scaffolds to enhance transport, medium perfusion to alleviate static diffusion limits, and carefully selected patient tissue sources—continue to advance the feasibility of this approach for both research and eventual therapy.

To crystallize the merits and drawbacks of organotypic testis culture, we provide a SWOT analysis in Table 1.

### 3.2. Testicular Organoids and 3D Bioengineered Systems

Testicular organoids and 3D bioengineered systems represent efforts to recreate the complex architecture and microenvironment of the testis through self-organizing or scaffold-supported structures [7,40]. These platforms aim to improve upon organotypic slice culture by providing enhanced spatial organization, nutrient diffusion, and long-term support for germ–somatic cell interactions.

Dissociated human testicular cells can self-assemble into organoids that partially recreate seminiferous microarchitecture. Early microwell-aggregated organoids contained Sertoli, Leydig, and peritubular-myoid cell populations but displayed minimal germ cell maturation [41]. A pivotal advance was the reconstruction of basement membrane-delimited, tubule-like domains from prepubertal cells, establishing, for the first time, an in vitro separation of basal and adluminal compartments, though again without completion of spermiogenesis [42]. Spatial fidelity has since improved through scaffold-assisted approaches. Collagen- or alginate-based hydrogels have supported haploid differentiation (≤17.9% round spermatids) in tissue from men with obstructive azoospermia or maturation arrest [43,44]. However, chromosomal errors remain a concern: XY-double-positive spermatids were detected in Matrigel cultures, underscoring the need for rigorous aneuploidy surveillance before any clinical application [45].

Three-dimensional bioprinting now represents the cutting edge of architectural control. Robinson et al. printed patient-derived testicular cells using an alginate–collagen bio-ink to create hollow tubules. Within 12 days, meiotic (*SYCP3*) and post-meiotic (*TNP1*) transcripts were up-regulated, demonstrating that geometry alone can enhance differentiation [46]. In a separate study, platelet-rich plasma scaffolds enriched undifferentiated SSCs two- to four-fold compared to two-dimensional culture, confirming the importance of biomechanical and paracrine cues [47]. Despite these advancements, most organoid systems still yield < 1% morphologically normal haploid cells and lack vascularization, limiting long-term viability. However, continued advances in biomaterial design, scaffold functionalization, and the integration of dynamic flow systems may enhance the accuracy of these models and bring them closer to replicating complete spermatogenesis in vitro. A concise SWOT analysis summarizing key aspects of testicular organoid and 3D bioengineered systems is presented in Table 2.

### 3.3. Microfluidics and Bioreactors

Dynamic perfusion systems—microfluidic “testis-on-chip” devices and macroscopic bioreactors—have emerged as promising tools to overcome the limitations of static culture by providing controlled fluid flow and shear stress [48,49]. By introducing continuous medium perfusion, these systems actively deliver nutrients and oxygen and remove wastes, more closely mimicking the interstitial fluid dynamics of the testis than static approaches [50]. Unlike simple organotypic slices, which develop steep nutrient gradients, microfluidic platforms maintain convective transport, alleviating hypoxia and nutrient depletion in the tissue [51]. A wide variety of testis-on-chip designs have been developed; generally, they consist of microchannel networks or chambers that permit precise regulation of media flow, factor gradients, and hormonal timing [52]. Some devices culture intact testicular fragments in microchannels under flow, while others seed isolated cells into interconnected microchambers—for example, one recent human testis-on-chip incorporated dual microfluidic chambers for Sertoli and Leydig cells, linked by perfused channels to enable reciprocal endocrine crosstalk, along with separate compartments for vascular endothelial cells and macrophages to recreate the complex multicellular testis environment [19]. This diversity of designs underscores that “testis-on-chip” is not a single technique but a versatile engineering framework, with each platform tailored to model different aspects of testis physiology.

Proof-of-concept studies in animals demonstrate the benefits of microfluidic perfusion. In rodent testes, microfluidic chips have sustained tissue viability and supported partial spermatogenic progression for extended periods, achieving more advanced differentiation than equivalent static cultures [53]. In porcine models, introducing gentle fluid flow doubled the yield of meiotic and post-meiotic germ cells compared to static ALI culture, highlighting perfusion as a key determinant of efficiency [54]. To date, no microfluidic system has achieved complete spermatogenesis with fully mature sperm in human or non-human primate cultures. Nevertheless, progress is evident: a recent polydimethylsiloxane (PDMS)-based microchip maintained small fragments of immature human seminiferous tubule for over 14 days while preserving tubular architecture and sustaining endocrine function (stable testosterone and inhibin B output) under pulsatile FSH/luteinizing hormone (LH) input. Notably, this platform maintained populations of haploid (1n) and tetraploid (4n) germ cells throughout the culture—evidence that early meiotic events were preserved ex vivo [48,55]. These results, though falling short of full spermatogenesis, confirm that human germ cells can at least enter meiosis in a microfluidic environment, an encouraging step toward eventual in vitro spermiogenesis.

Bioreactors provide an analogous perfusion advantage on a larger scale. Flow-enabled bioreactor systems (e.g., spinner flasks, rotating wall vessles, and mini bioreactors) allow cultivation of larger tissue pieces or cells aggregates with improved nutrient exchange. They often incorporate 3D scaffolds or beads on which testicular cells can attach and reorganize. For instance, spinner flask culture has been applied to organoids derived from human pluripotent stem cell (PSC): when human embryonic stem cells (hESCs) were differentiated into proto-testicular organoids in a spinning bioreactor, the organoids exhibited key somatic cell markers (WT1 Sertoli-like cells, GATA4, and SOX9 cells) within 40 days. However, germline markers (DAZL, OCT4) remained absent in that system, underscoring that additional cues are needed for germ cell differentiation [56]. Current limitations of perfused bioreactors include tissue heterogeneity (pieces of tissue may not all receive equal flow) and shear stress that can damage cells if not carefully controlled.

Looking forward, ongoing engineering innovations aim to further enhance these dynamic culture platforms. “Hybrid” systems are being developed that integrate microfluidic microvasculature channels within 3D tissue scaffolds, providing both perfusion and a more biomimetic structural support. In addition, combining perfusion devices with co-culture strategies (e.g., adding peritubular myoid cells, macrophages, or MSCs into the circuit) is an active area of research, since recapitulating the full niche may synergistically improve germ cell development. With refinements in microarchitecture design, bioactive scaffolds, and precise flow control, microfluidic chips and bioreactors are expected to play an increasingly central role in achieving complete IVS. These platforms not only offer better control over the spermatogenesis microenvironment but also provide valuable testbeds for experimental perturbations (hormone addition schedules, toxicology assays, etc.), thereby accelerating the translation of IVS from bench to bedside [48]. Key SWOT elements of these perfusion-based systems are outlined in Table 3.

### 3.4. Somatic Cell Co-Culture Strategies

Somatic cell co-culture strategies aim to replicate key aspects of the testicular niche by combining germ cells with one or more supportive somatic cell types, most commonly Sertoli cells, peritubular myoid cells, or combinations of testicular stromal cells. These models seek to provide the paracrine, juxtacrine, and structural cues essential for germ cell survival, proliferation, and differentiation.

Early co-culture systems demonstrated that Sertoli cells could support the survival and limited differentiation of isolated spermatogonia or primordial germ cells in vitro, largely through the secretion of growth factors such as GDNF, and FGF2 [18,57]. Classic Sertoli cell feeder layers combined with recombinant FSH or testosterone have been shown to accelerate secondary spermatocyte and spermatid formation but often produce a shortened spermiogenesis window (≈24 h) and ≈12% morphological abnormalities [58,59,60]. Regulatory concerns regarding non-human feeder cells, such as Vero lines, have limited their translational potential, favoring human-derived or stem cell-based feeder systems [61,62]. Estradiol-primed fetal Sertoli cells have been reported to double the proportion of SSEA4^+^/c-KIT^+^ SSCs within one week, likely via enhanced SCF secretion [63]. Mesenchymal stem cell (MSC) co-culture prolongs tissue viability through the secretion of GDNF, vascular endothelial growth factor (VEGF), and thrombopoietin (TPO); the addition of MSCs to ALI systems increased SCP3^+^/acrosin^+^ spermatids and extended culture longevity to 28 days [64]. Recent studies using soft-agarose laminin matrices embedding adult SSCs with Sertoli cells have reported the formation of morphologically mature sperm-like cells after ≈ 60 days, although fertilization competence has not yet been demonstrated [65,66].

Key challenges of this methodology include achieving physiologically relevant ratios and spatial organization of germ and somatic cells, replicating the dynamic signaling changes required at each stage of spermatogenesis, and preventing somatic cell overgrowth or dedifferentiation during prolonged culture. A persistent hurdle is also the intrinsic dysfunction of Sertoli cells in patients with NOA, which is characterized by defective *GDNF* and *SCF* expression, as well as impaired tight-junction formation within the blood–testis barrier [67,68,69,70]. Therefore, gene editing or reprogramming of autologous somatic cells is considered a priority for the development of xeno-free clinical protocols. The main SWOT considerations of somatic cell co-culture approaches are outlined in Table 4.

### 3.5. Growth Factor-Driven Differentiation Protocols

Growth factor-driven differentiation protocols represent a reductionist approach aimed at directing germ cell development by supplying exogenous signaling molecules to isolated germ cells or pluripotent stem cell-derived progenitors. These methods seek to replicate the stage-specific cues of the testicular niche through defined combinations and temporal delivery of growth factors, cytokines, and small molecules in two- or three-dimensional culture systems.

GDNF and basic fibroblast growth factor (bFGF) are universally recognized as indispensable for SSC self-renewal, preventing apoptosis and upregulating canonical stemness markers, such as PLZF, GFRα1, and ID4 [33,71]. SCF acts downstream to support differentiating spermatogonia, and its bioavailability can be boosted indirectly via β-estradiol stimulation of Sertoli cells [63]. RA serves as the canonical trigger for meiotic initiation through STRA8 activation. However, its effect is age- and context-dependent: RA supports meiosis in infant tissue but may induce degeneration in some fetal cultures [72]. Consequently, sequential (two- or three-phase) protocols have been developed. These typically begin with an expansion phase rich in GDNF/bFGF/SCF, followed by a differentiation phase where RA, testosterone, and gonadotropins are introduced while self-renewal factors are tapered [73,74]. Bone morphogenetic proteins (BMP4/8) and Activin A further potentiate differentiation in 3D or PSC-derived systems, with their stage-specific dosing currently under active investigation [74,75]. Small molecule epigenetic modulators, such as valproic acid (a histone-deacetylase inhibitor) and vitamin C, have been shown to synergize with canonical growth factors to enhance chromatin remodeling and raise haploid yield to 2–5% in PSC platforms [71]. Similarly, feeder-free ALI cultures supplemented with concentrated MSC secretome (GDNF, VEGF, TPO) have been reported to prolong germ cell survival and meiotic progression [76].

Despite these incremental gains, haploid efficiency rarely exceeds 20%, and complete sperm maturation remains inconsistent. Future progress will depend on single-cell-resolved mapping of temporal signaling requirements, combined with real-time metabolic sensing in perfused bioreactors. This approach will pave the way for “smart” media that dynamically modulate factor concentrations in a stage-specific manner. The main SWOT considerations of growth factor-driven protocols are outlined in Table 5.

## 4. From Rodents to Humans: Key Achievements and Remaining Barriers

Over the past decade, rodent models—particularly mice—have served as the principal platform for advancing IVS and artificial gametogenesis research. These systems have achieved critical milestones, including complete spermatogenesis from neonatal organotypic cultures and partial gamete development from pluripotent stem cells (PSCs). While preliminary successes have also been reported in rats, large animals, non-human primates, and early human models, translating these achievements beyond rodents has proven to be considerably more challenging. However, building on the foundation of neonatal mouse organ cultures, newer methodologies—such as microfluidic “testis-on-chip” platforms, defined culture media, and tissue-engineering approaches—are now being developed to more accurately replicate the testicular niche across species. This section critically examines key rodent breakthroughs, contrasts them with the ongoing challenges in large animals and humans, and outlines the barriers that must be overcome for clinical translation.

### 4.1. Successes in Mouse: Foundations, Organ Culture Milestones, and Stem Cell-Derived Gametes

Murine studies have formed the foundation of IVS research, owing to their rapid reproductive cycle, genetic tractability, and experimental versatility. They have delivered the most significant breakthroughs to date, demonstrating that the complete spermatogenic process can be replicated outside the body under defined conditions.

Specifically, the landmark study by Sato et al. [36] achieved the first complete ex vivo spermatogenesis by culturing neonatal mouse testis fragments at an air–liquid interface. This system produced fertilization-competent spermatozoa that successfully generated healthy offspring via ICSI. Subsequent refinements replaced serum with defined supplements—including KnockOut Serum Replacement (KSR), antioxidants, and precise hormone cocktails—to sustain seminiferous tubule architecture for up to six months while minimizing oxidative damage [36,51,77]. Optimization of oxygen tension, nutrient delivery, and culture duration underscored the critical importance of mimicking in vivo dynamics and laid the groundwork for later IVS systems.

Building on organ culture models, microfluidic platforms and 3D bioengineered systems have further advanced the field. Recent pumpless “testis-on-chip” designs have sustained mouse SSCs for over 40 days using mesenchymal stem cell-conditioned perfusate, markedly improving germ cell differentiation compared to static cultures [78]. Three-dimensional scaffolds and hydrogels have contributed to maintaining long-term tissue architecture and enhancing fluid and gas exchange, moving closer to recreating a functional in vitro niche.

Pluripotent stem cell (PSC)-based approaches complement organ culture by enabling de novo gamete generation. Mouse embryonic stem cells (ESCs) and induced PSCs (iPSCs) have been directed to form primordial germ cell-like cells (PGCLCs), which can complete meiosis under optimized feeder or organ culture conditions and give rise to spermatid-like cells [79]. Remarkably, these in vitro-derived gametes have produced viable, fertile offspring and have even enabled bimaternal and bipaternal mouse pups, demonstrating full karyotypic and epigenetic competence [78,79].

Collectively, these platforms have provided invaluable insights into the cellular, molecular, and engineering requirements for IVG, and serve as a foundation for ongoing efforts to translate these findings to larger animal and human models.

### 4.2. IVS in Rats and Large-Animal Models (Livestock and Non-Human Primates)

While rodent studies—especially in mice—have delivered landmark achievements in IVS, translating these advances to rats, livestock species, and non-human primates has proven considerably more difficult. Despite anatomical similarities, differences in spermatogenic cycle length, testicular architecture, and niche signaling contribute to these challenges.

At first, attempts to adapt murine organotypic culture protocols to rats resulted in only partial germ cell progression, despite careful control of oxygen tension, antioxidant supplementation, and growth factor delivery [80,81]. Most studies demonstrated that rat IVS progressed no further than the pachytene spermatocyte stage [82,83,84]. Complete spermatogenesis remained elusive until 2023, when Matsumura et al. combined metabolomics-guided media formulation with a transgenic germ cell reporter to support the formation of round spermatids in cultured rat testis fragments. Remarkably, ICSI using these IVS-derived spermatids produced healthy rat offspring, marking the first demonstration of full IVS-derived fertility in rodent species beyond mice [85].

In pigs and other livestock models, IVS efforts have largely focused on tissue maintenance, organoid generation, and developmental studies. Porcine testis fragments can retain seminiferous architecture and early germ cells for several weeks in defined media, but progression beyond early meiotic stages has not been achieved [86,87]. In bovine models, a 2015 study employed bovine testis tissue fragments cultured at 37 °C in mouse serum-free medium, supplemented with triiodothyronine (T_3_) and SCF. After three months, spermatogenesis was enhanced by 2.4–2.7-fold, demonstrating a significant induction of bovine SSCs in vitro [88]. In general, decellularized matrix-based organoids in livestock species have successfully reconstituted somatic compartments and vascular-like channels, offering valuable platforms for developmental research and toxicity testing, though functional sperm production remains out of reach.

Regarding non-human primates (NHPs), these models provide the closest physiological similarity to humans and serve as a critical preclinical bridge for IVS research. Organotypic cultures of marmoset testis fragments, maintained in microfluidic devices, have demonstrated sustained viability, tubular structure, and steroidogenesis, suggesting that dynamic perfusion may eventually support more advanced germ cell development [50,78]. Furthermore, although true IVS has not yet been achieved in NHPs, in vivo grafting of cryopreserved rhesus macaque testis tissue into host animals has successfully restored spermatogenesis following chemotherapy. Graft-derived sperm have been shown to produce healthy offspring through natural mating or assisted fertilization [89]. While grafting circumvents in vitro culture, it validates the concept of reactivating primate spermatogenesis and underscores the importance of ethically expanding NHP IVS research. Finally, testis-on-chip platforms and bioreactors applied in primate models remain at an early experimental stage, showing incremental gains in culture longevity and germ cell marker expression, but there have been no reports of functional sperm production to date.

### 4.3. Human IVS Efforts: Progress and Limitations

Human IVS efforts have heavily relied on lessons learned from rodent and primate models, but they are still in an early, exploratory stage. The complexity of human spermatogenesis, the long cycle length (~74 days), and limited access to suitable testis tissue have all posed significant challenges.

#### 4.3.1. Organ Culture of Fetal and Adult Tissue

Organ culture of human fetal and adult testis tissue represents the most direct attempt to replicate spermatogenesis in vitro. Organotypic cultures of second-trimester human fetal testis fragments have demonstrated partial meiotic entry and formation of haploid cells, but with low efficiency and no demonstration of functional competence [32,33,39]. Adult human testis biopsies, especially from patients with NOA, often experience rapid germ cell attrition and somatic cell senescence during culture. While there are sporadic reports of round spermatid formation in hydrogel or bioreactor-supported systems, these findings have yet to be independently validated [38]. Recent work with ALI and microfluidic platforms has modestly prolonged tissue viability and preserved endocrine function (e.g., testosterone, inhibin B secretion), but no mature, functional spermatozoa have yet been produced.

#### 4.3.2. Pluripotent Stem Cell and Organoid Platforms

Pluripotent stem cell-based and organoid platforms represent reductionist and bioengineered approaches that seek to generate germ cells de novo or to reconstruct key elements of the testicular niche. Human PSCs, including embryonic stem cells (ESCs) and iPSCs, can be directed to PGCLCs. Under feeder-supported or organoid conditions, these have yielded occasional haploid cells expressing markers such as acrosin or *TNP1*, though with inconsistent efficiency and incomplete epigenetic reprogramming [33,67]. Organoids constructed from dissociated human germ and somatic cells preserve the three-dimensional architecture and cell–cell interactions, but the formation of late spermatids remains sporadic and inefficient [38,78].

#### 4.3.3. Persistent Barriers and Translational Hurdles

Despite significant experimental advances, human IVS faces persistent biological and technical barriers that currently hinder its translation into clinical practice. Among the most critical barriers are as follows:

Somatic Support and Microenvironmental Complexity: Recreating the niche that sustains germline progression remains the foremost challenge, regardless of species. Sertoli cells provide structural, nutritional, and paracrine support. However, in patients with NOA, these cells often exhibit altered gene expression profiles, such as reduced *GDNF* or *SCF* expression, which compromises germ cell maintenance [68,69,70,90]. Even in healthy tissues, Sertoli cells tend to dedifferentiate or senesce during prolonged culture. Moreover, recent small-RNA profiling studies in testicular tissue from men with non-obstructive azoospermia have revealed distinct microRNA signatures associated with the failure of spermatogenesis and unsuccessful sperm retrieval. These molecular patterns may not only reflect intrinsic testicular dysfunction but also serve as potential markers to predict the feasibility of in vitro spermatogenesis in patient-derived samples [91]. Microfluidic methods that mimic microvasculature and supply exogenous growth factors, as tested in mouse [51,77,78] and primate models [50], may help circumvent these limitations, but they are not yet optimized for human IVS.

Meiotic Control and Epigenetic Reprogramming: A complete IVS protocol must reliably guide germ cells through meiosis, a stringent and highly regulated process that involves homologous chromosome pairing, crossing over, and epigenetic resetting. In vitro-derived spermatids often exhibit incomplete chromatin remodeling, asynchronous protamine replacement, or imprinting defects [32], which can jeopardize embryo viability and offspring health. While rodent IVS systems have produced healthy offspring under optimized conditions, human germ cells appear more vulnerable to culture stress and epigenetic errors [79,85]. Overcoming this challenge will likely require extended culture durations that align with the natural ~64–74-day spermatogenic cycle and precise timing of developmental cues, such as RA pulses.

Standardized Validation and Ethical Constraints: The clinical translation of IVS relies on rigorous, standardized validation. In rodents, the gold-standard test for functionality is fertilization and the generation of viable offspring [36,85]. However, human studies encounter legal and ethical limitations that prevent the generation of embryos solely for experimental validation. As a result, researchers depend on surrogate markers, such as haploid DNA content, acrosome formation, and gene expression profiles. Although these markers suggest the potential for successful spermatogenesis, they do not guarantee developmental competence [33,38,39]. No consensus exists on defining a “fully functional” sperm in vitro for clinical purposes. Moreover, scaling up any validated system to produce sufficient quantities of sperm under Good Manufacturing Practice (GMP) conditions introduces further logistical and regulatory challenges.

Safety Considerations: The final layer of caution lies in safeguarding the genetic and epigenetic integrity of in vitro-derived gametes. Long-term cultures or repeated passaging may accumulate mutations or imprinting errors that could be transmitted to offspring. For instance, while rodent models show that prolonged culture can maintain stable gametes under carefully controlled conditions [51,77,78,85], this level of stability has not yet been definitively proven for human cells. Emerging tools, such as single-cell technologies and epigenetic screens, show promise but standardized protocols for monitoring or mitigating these risks have not yet been established.

In summary, while the rodent paradigm establishes that IVS and IVG can produce functional gametes, human applications lag behind. The need for improved culture microenvironments, extended maintenance of spermatogenesis, and strict quality control of cell outputs frames the research agenda moving forward.

## 5. Pluripotent Stem Cells and Reprogramming Approaches

PSCs, encompassing hESCs and iPSCs, have emerged as a promising avenue for IVS research and potential clinical applications. These cells can differentiate into virtually any tissue type and thus represent a powerful platform to generate male germ cells in scenarios where no endogenous SSCs are available [e.g., Sertoli cell-only (SCO) syndrome], or in individuals rendered infertile by gonadotoxic treatments without prior fertility preservation [92,93,94]. This pluripotent stem cell-based approach directly aligns with the overarching goal of artificial gametogenesis: to create patient-specific gametes for those who lack functional sperm, complementing the organotypic IVS strategies described earlier. The key milestones that underpin the evolution of PSC-based IVG platforms are summarized in Figure 2.

Early milestones include the establishment of mouse embryonic stem cells (1981) and human ESCs (1998) [95,96]. By 2003–2011, foundational steps in deriving PGCLCs from pluripotent stem cells were achieved [97,98,99,100,101,102,103]. In 2011, mouse PGCLCs transplanted into testes yielded functional sperm that sired offspring [104]. Fully in vitro generation of haploid spermatids from mouse ESCs was reported in 2016, and the first complete in vitro oogenesis from mouse PSCs in the same year [105,106]. By 2018, human iPSCs had been induced to form oogonia-like cells in long-term culture (though not yet functional oocytes) [107]. In the same year, spermatid-like cells derived from human SSCs were shown to be functionally competent, achieving fertilization in vitro [108]. In 2021, spermatid-like cells differentiated from rhesus ESCs successfully fertilized primate oocytes and developed into blastocysts, marking a crucial milestone in non-human primate in vitro spermatogenesis [109]. Subsequently, in 2022, a microfluidic “testis-on-chip” platform was developed to support long-term ex vivo culture of primate seminiferous tubules with stable hormonal function and germ cell maintenance [110]. These milestones illustrate the rapid evolution of artificial gametogenesis, predominantly in the mouse model, and set the stage for ongoing advances in higher primates and humans. This section reviews landmark achievements in PSC-based germ cell differentiation, compares their efficacy, and discusses the prospects and challenges for clinical application.

### 5.1. Landmark Advances in Reprogramming and PSC Differentiation

The advent of induced pluripotent stem cell (iPSC) technology in 2007 enabled the reprogramming of adult somatic cells into PSCs through the introduction of defined transcription factors, including OCT4, SOX2, KLF4, and c-MYC [99]. Human iPSCs (hiPSCs) share key features with hESCs, such as the expression of canonical pluripotency markers (e.g., *OCT4*, *NANOG*, *SOX2*) and the capacity for indefinite self-renewal under appropriate culture conditions [111]. Because they can theoretically give rise to any cell type, hiPSCs open the door to personalized fertility therapies; for instance, somatic cells from an infertile patient can be reprogrammed and then differentiated into germline cells for subsequent use in assisted reproduction.

Early studies provided proof-of-concept that both hESCs and hiPSCs could be directed toward the male germ cell lineage. Initial efforts to produce germ cell-like cells (GCLCs) demonstrated that short-term culture in monolayer systems, supplemented with media such as alpha Minimum Essential Medium (αMEM), yielded cells expressing hallmark germline markers such as VASA (also known as DDX4), an RNA helicase specific to germ cells, and DAZL (Deleted in Azoospermia-Like), a key regulator of germ cell development, including a subpopulation of round spermatid-like cells [33]. Although these cells did not survive beyond 20 days, they highlighted the intrinsic potential of PSCs for partial spermatogenic differentiation in vitro.

Subsequent refinements aimed to enhance differentiation in later stages. Eguizabal et al. (2011) reported that hiPSCs are capable of completing meiosis and generating haploid post-meiotic cells without the forced overexpression of germline-specific genes [104]. Similarly, Easley et al. reported that hESCs and hiPSCs can produce a spectrum of germ cell-like populations, from SSC-like cells to spermatid-like cells expressing transition protein-1 (*TNP1*) and protamine-1 (*PRM1*), indicating partial maturation [33]. Such findings validate the feasibility of driving PSCs beyond primitive germ cell states.

Regarding animal studies, landmark protocols in mice combined BMP4, SCF, LIF, and other key signaling molecules to derive primordial germ cell-like cells (PGCLCs) from embryonic stem cells (ESCs) and iPSCs. When PGCLCs were aggregated with fetal gonadal somatic cells, they were able to complete spermatogenesis and produce viable offspring [103]. Inspired by these models, human PSC studies developed comparable directed differentiation strategies, yielding PGCLCs marked by *SOX17*, *BLIMP1*, and *TFAP2C* [112].

Together, these advances highlight the potential of PSCs in reproductive and regenerative medicine, while highlighting the considerable species-specific barriers that remain for human application.

### 5.2. Comparative Efficacy and Epigenetic Considerations

hESCs and human iPSCs (hiPSCs) share core properties. However, despite these similarities, hESCs and hiPSCs may exhibit subtle differences in their propensity to differentiate into germ cells. Some research groups have reported comparable efficiencies in producing PGCLCs when co-cultured with gonadal feeder layers, while others observed slightly higher yields from hiPSCs under specific conditions [101]. These disparities may reflect line-to-line variation rather than a fundamental difference between hESCs and hiPSCs. Although studies have demonstrated that hiPSCs derived from different somatic sources (e.g., fibroblasts, blood cells) can generate germ cell-like populations and, in some cases, haploid cells, the extent to which donor origin influences germline reprogramming potential in humans remains unclear. Overall, direct comparative analyses between hESCs and hiPSCs for IVG are limited, and further studies are needed to clarify their relative efficiency.

A more prominent concern involves epigenetic fidelity. hESCs are derived from the inner cell mass of blastocyst-stage embryos [113], whereas hiPSCs are reprogrammed from somatic cells (e.g., fibroblasts, blood cells) through the introduction of defined transcription factors [98]. This difference in origin imparts distinct molecular and epigenetic signatures. Germ cell development entails extensive methylation reprogramming: first, imprints are erased in primordial germ cells, and then paternal-specific imprints are re-established during spermatogenesis [32,114]. While both hESC- and hiPSC-derived germ cells initiate imprint erasure, incomplete demethylation has been reported in some hiPSC-based systems [115]. This suboptimal reset suggests that reprogramming methods or prior epigenetic ‘memory’ could impair certain hiPSC lines’ ability to fully mimic in vivo germline methylation. Encouragingly, later-stage PSC-derived spermatids in some protocols displayed typical paternal imprints at key loci (e.g., *H19*, *IGF2*), indicating that correct imprint establishment is achievable under refined conditions [33]. Nonetheless, ensuring the comprehensive restoration of epigenetic marks remains vital before any clinical application for both hESCs and hiPSCs.

From a translational perspective, hiPSCs offer ethical and practical advantages, including the ability to create patient-specific lines for personalized therapies and disease modeling without the use of embryos. However, concerns regarding genetic and epigenetic stability, potential reprogramming-induced mutations, and variability in germline competency remain active areas of investigation.

### 5.3. Translational Promise, Barriers, and Outlook

PSC-based IVS holds promise for men with conditions such as SCO syndrome, where endogenous germ cells are absent [92]. By converting a patient’s fibroblasts or blood cells into hiPSCs and differentiating them into functional sperm, it may be possible to restore fertility. Similarly, cancer survivors who underwent gonadotoxic treatments—particularly in childhood—might benefit from hiPSC-derived germ cells if no pre-treatment tissue was cryopreserved [93,94]. In principle, hiPSC technology allows the creation of autologous germ cells for personalized reproductive therapies, avoiding the ethical controversies associated with embryo-derived stem cells. Thus, the possibility of generating autologous sperm from minimal somatic samples has sparked intense interest in the translational potential of IVS.

However, substantial hurdles must be overcome before PSC-derived gametes can enter clinical use. Foremost among these is ensuring that PSC-derived gametes are genetically and epigenetically normal. Even minor imprinting errors could result in early embryonic loss or imprinting disorders in offspring [114]. An intriguing but unresolved observation is that PSC-derived germ cells often progress through spermatogenic stages more rapidly than they do in vivo. While normal human spermatogenesis spans over two months, some in vitro systems detect post-meiotic cells in under three weeks [33,116]. This accelerated timeline may reflect the absence of regulatory checkpoints present in the seminiferous epithelium, along with supra-physiological concentrations of differentiation cues in culture. Although rapid maturation is experimentally advantageous, it also raises questions about whether essential quality control steps are compressed or bypassed. One critical functional requirement is the expression and activity of phospholipase C zeta (PLCζ), the sperm-specific factor that triggers oocyte activation via calcium oscillations. Kaltsas et al. underscored that deficiencies or mutations in PLCζ are linked to failed fertilization and have been identified in infertile men with otherwise normal sperm parameters. As such, any PSC-derived sperm intended for clinical use must demonstrate not only morphological maturity but also functional capacity for oocyte activation via PLCζ-mediated pathways [117].

Ongoing research is investigating whether these fast-tracked cells fully recapitulate normal sperm biology or if they harbor subtle molecular deficits that could compromise functionality or may be associated with genetic and epigenetic problems. Additionally, large-scale optimization is needed to yield sufficient numbers of mature spermatids or sperm, as current protocols often show low efficiency beyond the SSC-like or early meiotic stages. Regulatory and ethical considerations further complicate translation to the clinic, particularly given legal restrictions on generating embryos solely for testing in many jurisdictions.

Nevertheless, the field has progressed dramatically since the first hiPSC reprogramming studies [99]. Ongoing improvements in differentiation techniques, 3D testicular modeling, and epigenetic safeguarding strategies may ultimately make PSC-derived sperm a viable option for selected patient populations. In the interim, PSC-based IVS remains an invaluable research platform, shedding light on human spermatogenic mechanisms and enabling drug or toxicology screens relevant to male fertility. With continued refinement, along with ethical oversight and societal dialog on the acceptable boundaries of reproductive innovation, this technology may one day revolutionize infertility treatment for men lacking viable endogenous germ cells. Figure 3 presents an integrated development pipeline that maps these experimental breakthroughs onto the sequential steps required for future clinical implementation.

The following sections translate these technical advances into clinical, ethical, and future research perspectives.

## 6. Clinical Applications and Future Scenarios

Having explored the foundational technologies and experimental achievements, the review now turns to potential clinical applications of IVS and IVG. In the following subsections, we map how these emerging gametogenesis strategies could address specific causes of male infertility and novel family-building scenarios. By focusing on cases ranging from non-obstructive azoospermia to same-sex reproduction, we illustrate how artificial gametogenesis aligns with its central aim: offering fertility solutions for patients who currently have no viable options.

### 6.1. Non-Obstructive Azoospermia

NOA represents one of the most challenging categories of male infertility. Men in this group fail to produce mature sperm, often due to intrinsic testicular dysfunction [118]. Current treatment options for NOA are limited. Although procedures such as microdissection testicular sperm extraction (micro-TESE) can retrieve sperm in some NOA cases, success rates hover around 50%, leaving many patients with no option other than donor sperm [119]. In addition to TESE, varicocelectomy has been proposed as a potential intervention in selected NOA cases to enhance spermatogenesis. However, its efficacy remains controversial. As summarized by Kaltsas et al. [120], current evidence is inconclusive, with some studies reporting post-operative sperm appearance in the ejaculate, while others show minimal benefit. This ambiguity underscores the need for patient-specific stratification and further trials before varicocelectomy can be routinely recommended in NOA. The emergence of IVS offers a novel, potentially revolutionary solution. By culturing any residual SSCs from a patient’s testicular biopsy—or by reprogramming a somatic cell into an induced pluripotent stem cell (iPSC) and directing it along the spermatogenic pathway—it may be possible to generate haploid gametes fully in vitro [121].

Early-stage research has demonstrated the derivation of round spermatids from patient-derived SSCs in organoid and monolayer culture systems, and preliminary evidence from animal models indicates that these in vitro-created germ cells can fertilize oocytes and form early embryos [122]. Moreover, gene-editing tools such as CRISPR–Cas9 have been successfully used to correct known mutations (e.g., in the Kit gene) in murine SSCs, restoring fertility upon transplantation into sterile recipients [123]. Although these strategies remain distant from routine clinical use, they illustrate the growing potential of IVS to address the most challenging forms of male infertility.

### 6.2. Klinefelter Syndrome and Other Genetic Forms of Spermatogenic Failure

Klinefelter syndrome (47,XXY) is the most common sex chromosome aneuploidy in males, affecting approximately 1 in 500 live births, and is a major genetic cause of spermatogenic failure [124]. The condition is characterized by testicular atrophy, hyalinization of seminiferous tubules, Sertoli and Leydig cell dysfunction, and nearly universal azoospermia. While micro-TESE can occasionally retrieve sperm in mosaic cases or focal areas of spermatogenesis, the sperm retrieval rate is only 50% [125], leaving limited options for parenthood for these patients. Other genetic causes of spermatogenic failure include Y-chromosome microdeletions (e.g., AZFa, AZFb, AZFc deletions), mutations in genes regulating meiosis, and rare monogenic causes of Sertoli cell-only syndrome (SCO) or maturation arrest [126]. For these patients, conventional sperm retrieval techniques may fail, with donor sperm remaining the only reproductive option.

In all these cases, PSC-derived germ cells could, in theory, provide a new treatment avenue, especially if reprogrammed from autologous somatic cells. This approach could potentially bypass the intrinsic testicular failure associated with these conditions, as explained above.

### 6.3. Oncofertility: Fertility Preservation in Prepubertal Patients

Oncofertility focuses on preserving the reproductive potential of individuals affected by cancer and its treatments [127]. Unlike postpubertal males, who can bank sperm prior to treatment, prepubertal boys lack mature sperm and thus face permanent infertility if their testicular tissue is destroyed by chemotherapy or radiation [128]. Experimental protocols involving cryopreservation of immature testicular tissue (ITT) have been implemented in pediatric settings worldwide, with the long-term goal of either transplanting this tissue back after cancer remission or using it to produce sperm in vitro [129].

Numerous organotypic culture approaches, including air–liquid interface setups and three-dimensional scaffolds, have demonstrated partial progression of spermatogenesis in tissue from infant or fetal testicular biopsies [130]. Notably, researchers have reported the development of haploid round spermatids in vitro from human fetal testicular tissue, as well as successful fertilization and early embryonic development using such cells in preclinical models [39]. Although no human births have occurred via this approach yet, similar methods have resulted in live offspring in non-human primates following the autografting of cryopreserved testicular fragments [131].

Furthermore, for patients without preserved tissue, or when the tissue is unavailable or unusable, PSC-derived germ cells may provide a unique solution. Autologous iPSCs, generated from the patient’s somatic cells, could theoretically be differentiated into functional spermatozoa for assisted reproduction. This approach bypasses the need for prior gamete or tissue banking; however, it remains experimental and requires extensive validation before clinical application. Overall, once safety and efficacy are confirmed, IVS-based oncofertility solutions could enable pediatric cancer survivors to achieve genetic parenthood.

### 6.4. Same-Sex Reproduction

One of the most intriguing—and ethically complex—dimensions of IVS research lies in the realm of same-sex reproduction. In principle, two men or two women could have a genetically related child by using cells from one partner to create “artificial” sperm or oocytes, which are then fertilized by gametes from the other partner [132]. This concept has been illuminated by mouse experiments demonstrating “bipaternal” and “bimaternal” offspring, although significant genetic and epigenetic manipulation is required to overcome imprinting barriers [133].

Recent studies in mice, for example, have taken male (XY) embryonic stem cells and induced them to lose the Y chromosome, producing an XX cell line capable of differentiating into oocytes. These oocytes, in turn, were fertilized by sperm from another male, resulting in viable pups [134]. Although these findings highlight the biological feasibility of same-sex reproduction, they also underscore the complexity involved, particularly in humans, where imprinting errors and long-term safety must be thoroughly addressed [135]. For now, this application remains speculative, but it represents a focal point for ethical debate and scientific exploration.

### 6.5. Differences of Sex Development

Individuals with differences of sex development (DSD) present with a wide spectrum of chromosomal, gonadal, or anatomical variants. Many forms of DSD involve dysfunctional or absent germ cells, often leading to infertility [136]. These conditions include, among others, 46, XY gonadal dysgenesis, ovotesticular DSD, and certain forms of androgen insensitivity or steroidogenesis defects [137]. Current options, such as the use of donor gametes, do not satisfy the desire for genetically related offspring in many affected individuals. IVS, however, could circumvent these limitations by generating gametes from reprogrammed iPSCs. In rare cases where dysgenetic gonadal tissue retains SSCs or precursor germ cells, such tissue could also serve as a starting point for in vitro maturation [138].

Early laboratory work has shown that iPSCs derived from patients with genetic causes of infertility (e.g., XX male syndrome) can be directed to differentiate into PGCLCs in vitro. By combining these iPSCs with supportive somatic cells in 3D scaffolds or organoid cultures, researchers are working toward recreating the microenvironment needed for advanced spermatogenic development [139]. Additionally, gene-editing technologies have demonstrated proof-of-concept for modifying sex chromosome content or correcting specific genetic anomalies—for example, strategies to eliminate an extra X chromosome—suggesting that one day it may be feasible to correct specific anomalies [140]. Although these approaches remain at an early, experimental stage, they indicate a future where individuals with DSD or related conditions might achieve autologous fertility through highly personalized, cell-based therapies [141].

### 6.6. Older and Solitary Individuals

Changing societal norms, along with biological realities, drive interest in extending fertility for older or solitary individuals. Men experience a gradual decline in sperm quality as they age, while women face a more abrupt depletion of ovarian reserve. IVS, and more broadly, IVG, could provide an alternative to donor gametes by enabling older adults to generate “younger” germ cells from their own somatic tissue [142]. A hypothetical scenario would involve a woman past menopause deriving iPSCs from her skin or blood cells and differentiating them into functional oocytes that might be fertilized through standard IVF. Similarly, men could use reprogrammed cells to create higher-quality sperm that is unaffected by age-related mutations [143].

Moreover, solitary individuals—who would otherwise require a donor to provide a missing gamete—may seek IVS as a solution for single-parent genetic reproduction. One person’s iPSCs could theoretically be directed to form both sperm and oocyte equivalents, although serious concerns about homozygosity and genetic risk would need to be addressed. While this scenario raises ethical and medical discussions, the technological path to achieving it significantly overlaps with ongoing IVS research and development [144].

### 6.7. A Precision Medicine Framework for IVS

The convergence of IVS with precision medicine offers one of the most exciting prospects for future clinical translation. By generating autologous gametes from patient-specific iPSCs, IVS could provide highly personalized fertility treatments for individuals with otherwise untreatable forms of spermatogenic failure. This approach would allow the alignment of fertility interventions with the unique genetic, epigenetic, and molecular profile of each patient. Specifically, by performing single-cell transcriptomics on testicular tissues, researchers can pinpoint specific molecular blockages in spermatogenesis (e.g., disrupted Wnt–β-catenin signaling) or examine the interplay of hormones and transcription factors. Such analyses could then guide the optimization of culture media and scaffold conditions to meet each patient’s unique requirements. A patient’s cultured germ cells, for example, might benefit from Wnt pathway modulators, RA supplementation, or immunomodulatory molecules tailored to correct underlying dysregulations [145].

In tandem, gene editing could address monogenic causes of infertility before cells ever undergo spermatogenesis in vitro. By correcting known mutations in SSCs or iPSCs using CRISPR-based methods and thoroughly screening for off-target effects, clinicians may eventually produce genetically “repaired” sperm that is both functional and free of certain inheritable disorders. Indeed, preclinical mouse studies have shown that CRISPR-edited SSCs can be transplanted back into infertile males, leading to successful sperm production and healthy offspring. Extending such breakthroughs to humans will depend on meticulous safety evaluations, regulatory oversight, and broad ethical consensus [142].

### 6.8. Research and Diagnostic Applications

In addition to their potential for restoring fertility, IVS systems offer powerful tools for advancing research and diagnostic approaches in male reproductive health. Patient-specific, iPSC-derived germ cells and testicular organoids provide unique models to study the molecular mechanisms underlying spermatogenic failure, including the role of genetic mutations, epigenetic dysregulation, and somatic niche dysfunction [146,147]. These platforms enable the investigation of conditions such as NOA, Klinefelter syndrome, and disorders of sex development in a controlled, human-specific setting that overcomes the limitations of animal models. IVS and organoid models also hold promise for toxicology studies and drug screening. By recreating human spermatogenesis in the laboratory, these systems could be used to assess the gonadotoxic potential of chemotherapeutic agents, environmental toxins, or novel pharmaceuticals in a patient-specific manner [146]. Such applications could support personalized risk assessment for fertility preservation, particularly for cancer patients or individuals with occupational exposures. Finally, as IVS technologies evolve, they may contribute to diagnostic workflows by providing functional assays of spermatogenic potential in patients with ambiguous testicular histology or unclear prognosis, complementing existing genetic and endocrine assessments. However, these applications remain experimental and require further validation before clinical implementation.

## 7. Future Directions and Research Gaps

Recent advances in IVS and artificial gamete generation have fueled optimism that novel fertility-restoration strategies may soon become clinically viable [72,148]. However, critical gaps remain in our understanding of how to replicate the complexities of spermatogenesis in culture. This section outlines key directions for future research and highlights several conceptual, technical, and ethical obstacles that must be addressed to translate IVS from the laboratory to routine clinical practice [149]. Notable limitations include the fact that no in vitro gametogenesis system has yet produced fully functional human spermatozoa, the relatively low efficiency and incomplete maturation of gametes in current systems, and the risk of epigenetic aberrations or genetic mutations arising during extended culture. Additionally, significant ethical and translational challenges remain—from debates over germline genetic modification to the lack of established regulatory pathways for clinical use of lab-grown gametes. Ensuring clinical viability is indeed the unifying objective driving research in artificial gametogenesis, underscoring that addressing these scientific and societal challenges is essential for ultimately delivering a safe and effective infertility treatment.

Enhancing Efficiency and Completeness of Gametogenesis: A key priority for advancing IVS is the development of culture systems capable of supporting the entire human spermatogenic cycle, from SSCs to mature, functional spermatozoa. However, a consistent challenge is the relatively low yield of fully differentiated, haploid germ cells. In current human systems, fewer than five percent of cultured cells typically progress to haploid gametes, underscoring the need to improve both the efficiency and fidelity of in vitro differentiation [72,148]. Potential strategies include high-throughput screening of growth factors, small molecules, and culture conditions that promote meiosis and late-stage spermiogenesis. Advanced systems biology approaches—integrating transcriptomic, proteomic, and metabolomic data—may reveal critical pathways missing in vitro [150,151]. Additionally, evidence suggests that physical cues, such as fluid flow and shear stress, along with hormonal stimuli (e.g., thyroid hormones [37] known to influence spermiogenesis in vivo), could further enhance the transition from round spermatids to morphologically mature sperm [35,43].

Developing Advanced Biomimetic Culture Systems: Reproducing the highly specialized microenvironment of the human testis in vitro remains one of the greatest challenges for achieving complete spermatogenesis. The seminiferous tubules are intricate, three-dimensional structures that coordinate germ cell progression through precise spatial organization, dynamic paracrine signaling, and stage-dependent endocrine inputs [34]. One promising approach involves organ-on-chip devices designed to mimic seminiferous tubules in parallel microchannels, allowing for the co-culture of Sertoli cells and germ cells under precisely managed fluid dynamics. Additionally, 3D bioprinting techniques have the potential to recapitulate the tubular geometry of native testicular tissue. By utilizing “smart” bio-inks and microsphere-based drug delivery systems, researchers can release signaling molecules in a controlled manner that more closely matches the temporal regulation observed in vivo [152,153]. Such innovations could be particularly valuable for addressing conditions like NOA, where tissue architecture and cell–cell interactions are significantly compromised.

Ensuring Genetic and Epigenetic Integrity: A major challenge for IVS is ensuring that the derived germ cells possess correct genetic and epigenetic programming. Future studies must focus on developing robust methods to assess the genetic and epigenetic fidelity of IVS-derived gametes. This includes applying single-cell genomics, methylome profiling, and chromatin accessibility assays to validate imprinting status, chromosomal integrity, and transcriptional competence. Particular attention should be given to sex chromosome dynamics, such as X-chromosome inactivation and meiotic sex chromosome inactivation, both of which are essential for normal spermatogenesis [154]. Successfully replicating these processes in vitro will be a major step toward producing gametes that are functionally equivalent to those generated in vivo.

Validation in Animal Models and Safety: Before proceeding to human clinical trials, it is essential to validate IVS techniques in non-human primates or other large animal models to demonstrate both the functional competence and the safety of the derived gametes. For example, autologous induced iPSCs derived from a macaque could be differentiated into sperm in vitro and used for IVF, followed by embryo transfer to a surrogate monkey to assess both efficacy and safety. These studies would provide a more rigorous evaluation of in vitro sperm functionality than rodent experiments and could help refine culture conditions while minimizing the ethical risks associated with premature human trials [122]. Comprehensive long-term studies are also needed to monitor the health, fertility, and transgenerational effects of IVS-derived gametes in animal models before human clinical trials can be ethically justified.

Establishing a Clinical Translation Pipeline: Even as scientific breakthroughs accumulate, significant challenges remain in developing a standardized pipeline for clinical application. For instance, obtaining and expanding human SSCs from small testicular biopsies may be challenging, and generating high-quality iPSC lines under GMP conditions necessitates improved reprogramming techniques to reduce variability [122]. Additionally, the prolonged culture times and complex, multi-step protocols commonly associated with IVS are difficult to incorporate into feasible clinical workflows. Automated culture systems equipped with sensors for real-time feedback on cell growth and differentiation may help address these logistical and quality control issues while decreasing the risk of contamination and operator error [155].

Integration of Gene Editing and Genome Engineering: The integration of gene-editing and genome engineering technologies offers exciting possibilities for advancing IVS, particularly for individuals with monogenic causes of infertility. In conditions such as Y-chromosome microdeletions, *TEX11* mutations, or specific disorders of sex development, gene correction at the pluripotent stem cell (PSC) stage could enable the production of genetically intact gametes from autologous cells. Preclinical studies have demonstrated the feasibility of precise genome editing in PSCs using CRISPR-Cas9 [156]. However, combining gene editing with IVS poses significant challenges, including the need for stringent validation of editing precision, avoidance of off-target effects, and long-term monitoring of genetic and epigenetic stability. Ethical concerns regarding germline modification further complicate the path toward clinical translation, necessitating careful regulatory oversight and societal dialog before implementation.

Ethical, Social, and Regulatory Considerations: The potential application of IVS and related technologies raises profound ethical, social, and regulatory questions that must be addressed alongside scientific progress. Public acceptance of IVS will depend on transparent communication of its risks and benefits, as well as broad societal dialog about the moral status of laboratory-derived gametes and embryos. Key concerns include the potential for unintended genetic or epigenetic alterations, transgenerational risks, and the broader implications of producing offspring using gametes derived through reprogramming or genome editing. These concerns are amplified when IVS is combined with emerging gene-editing technologies. Concurrent research in bioethics and the social sciences will be crucial to understanding patient perspectives, informing public debate, and shaping consent processes—especially in pediatric contexts where testicular tissue might be biobanked for future use [39]. From a regulatory standpoint, international guidelines specific to IVS are currently lacking. Regulatory oversight varies widely across countries regarding embryo research, germline intervention, and stem cell-based technologies. Developing clear, harmonized regulatory frameworks will be essential to ensure responsible innovation, safeguard patient interests, and define boundaries for permissible research. International collaboration among professional societies, regulators, and stakeholders could facilitate data sharing, establish uniform safety standards, and build public trust in this emerging field.

## 8. Conclusions

IVS and artificial gametes constitute a cutting-edge avenue in reproductive biology, opening the door to unprecedented solutions for male infertility. Recent advancements in organotypic culture, microfluidic “testis-on-chip” devices, 3D organoids, and stem cell reprogramming have collectively demonstrated that the molecular and cellular architecture of spermatogenesis can be partially recapitulated ex vivo—particularly in rodent models, where complete maturation of functional sperm and the birth of healthy offspring have been achieved. Translating these findings to humans faces substantial hurdles: sensitive germ cells require a precisely orchestrated somatic niche, including accurate hormonal pulses, intact testicular architecture, and proper epigenetic remodeling over a ~64–74-day maturation process. While pilot human organoid and culture systems have produced round spermatid-like cells, a fully consistent protocol for creating mature, functional sperm remains elusive. Nevertheless, the field has progressed significantly by identifying key signaling pathways (e.g., GDNF, SCF, and RA), pioneering organoid and perfusion-based approaches, and refining pluripotent stem cell methods that may one day enable patients with NOA, survivors of gonadotoxic therapies, or even same-sex couples to have genetically related offspring.

No in vitro system has yet produced fully functional human sperm, but converging progress in 3D bioprinting, microfluidic perfusion, and multi-omics quality control outlines a concrete translational roadmap. Looking ahead, several avenues promise to enhance the safety and efficiency of IVG. The development of biomimetic culture systems—incorporating 3D bioprinting, microfluidic perfusion, and engineered somatic support cells—aims to meet the multifaceted demands of germ cell maturation and reduce the high attrition rates currently observed in culture. Leveraging single-cell multi-omics, gene editing, and improved stem cell reprogramming methods could further elucidate the intricate processes of meiosis and epigenetic reprogramming while minimizing abnormalities that might compromise fertility or offspring health. Finally, ethical and regulatory considerations surrounding artificial reproduction underscore the importance of thorough clinical validation before widespread implementation. Although achieving full human IVS remains a major scientific and societal challenge, ongoing interdisciplinary research brings this goal within reach, heralding a new era for the treatment and study of male infertility.

## Figures and Tables

**Figure 1 ijms-26-07383-f001:**
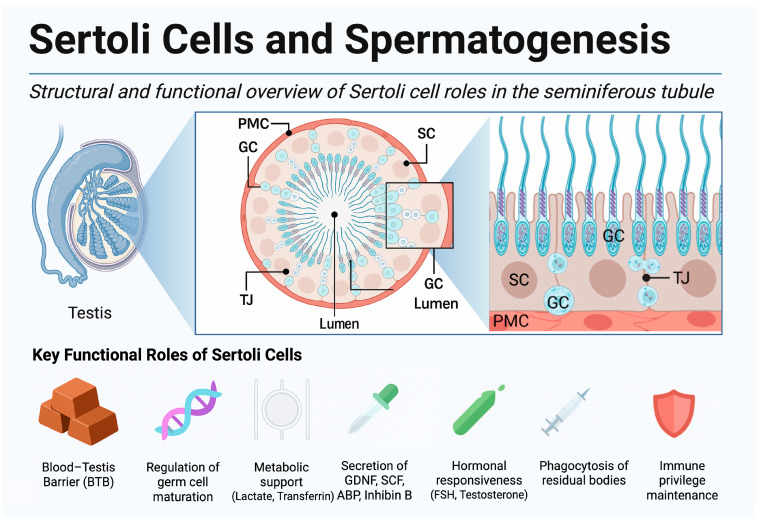
Anatomical localization and functional roles of Sertoli cells in the seminiferous tubule. Schematic overview of the testis and a representative seminiferous tubule shown in cross- and longitudinal views. SC extend from the basement membrane to the lumen, forming TJ that constitute the BTB. They nurture GC, secrete key trophic factors—GDNF, SCF, ABP, inhibin B—respond to FSH and testosterone, phagocytose residual bodies, and sustain immune privilege. The bottom panel summarizes these principal functions. Abbreviations: ABP, androgen-binding protein; BTB, blood–testis barrier; FSH, follicle-stimulating hormone; GC, germ cell; GDNF, glial cell line-derived neurotrophic factor; Inhibin B, inhibin beta; PMC, peritubular myoid cell; SC, Sertoli cell; SCF, stem cell factor; TJ, tight junction. Created in BioRender. Kaltsas, A. (2025) https://BioRender.com/1m0mbsy, accessed on 18 July 2025.

**Figure 2 ijms-26-07383-f002:**
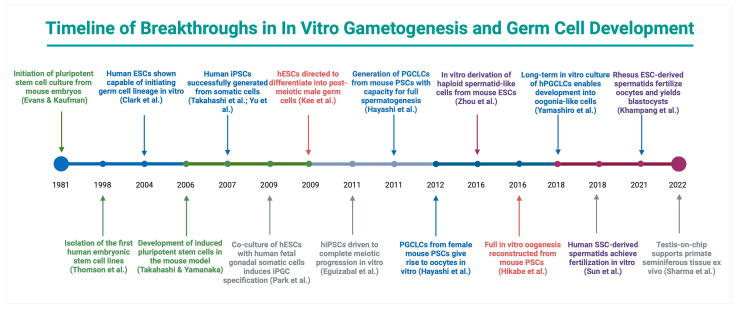
Milestones in Pluripotent Stem Cell Research and In Vitro Gametogenesis. Chronological overview of key experimental breakthroughs in the development of human and murine in vitro gametogenesis (IVG) platforms. Each event is color-coded according to its thematic focus, including pluripotent stem cell derivation, germline commitment, PGCLC differentiation, and maturation to functional haploid gametes. Created in BioRender. Kaltsas, A. (2025) https://BioRender.com/j9i2j9n, accessed on 18 July 2025 [95,96,97,98,99,100,101,102,103,104,105,106,107,108,109,110].

**Figure 3 ijms-26-07383-f003:**
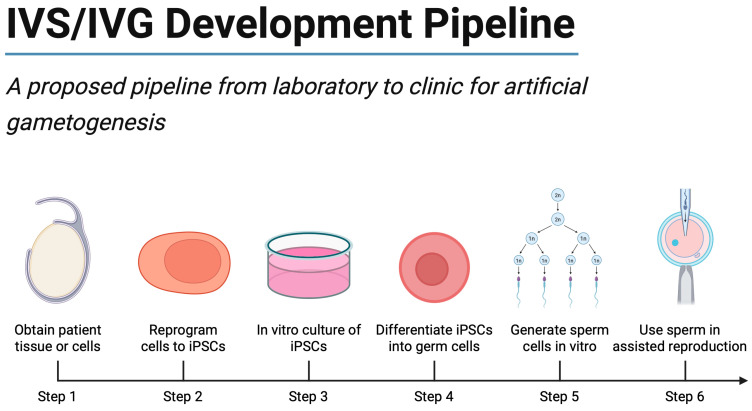
IVS/IVG Development Pipeline. A proposed pipeline from laboratory to clinic for artificial gametogenesis. This infographic illustrates the sequential steps from obtaining patient tissue or cells (e.g., testis biopsy or skin/blood for iPSC reprogramming), through in vitro culture and differentiation (including use of bioengineered scaffolds and growth factors), to the generation of sperm cells and their use in assisted reproduction (e.g., IVF/ICSI). This figure underscores the translational vision, integrating the advances reviewed into a cohesive roadmap. Created in BioRender. Kaltsas, A. (2025), https://BioRender.com/zsmhj1h, accessed on 18 July 2025.

**Table 1 ijms-26-07383-t001:** SWOT analysis of organotypic testis culture as an IVS platform.

Strengths	Weaknesses	Opportunities	Threats
Preserves native seminiferous tubule architecture Maintains authentic germ–somatic interactions Proven efficacy in rodent models (fertile offspring) Patient-specific tissue use avoids immunogenicity Suitable for toxicology and drug testing	Limited oxygen/nutrient diffusion in static systems Culture longevity insufficient for full human spermatogenesis Incomplete BTB reformation High SSC attrition; low haploid yield in humans Requires invasive biopsy; limited tissue availability	Integration of biomaterials (hydrogels, scaffolds, perfusion bioreactors) Ex vivo application of growth factors (e.g., GDNF, SCF) or gene editing Use in personalized oncofertility (e.g., cryopreserved prepubertal tissue) Benchmark for evaluating emerging IVS technologies	Tissue access limitations, especially in SCO or fibrotic testes High inter-patient variability in tissue response Risk of culture-induced abnormalities (e.g., epimutations) Rapid progress of alternative platforms (PSC-IVG, microfluidics) Ethical/regulatory constraints in human gamete derivation

Abbreviations: IVS, in vitro spermatogenesis; BTB, blood–testis barrier; SSC, spermatogonial stem cell; PSC, pluripotent stem cell; IVG, in vitro gametogenesis; SCO, Sertoli cell-only.

**Table 2 ijms-26-07383-t002:** SWOT analysis of testicular organoids and 3D bioengineered systems.

Strengths	Weaknesses	Opportunities	Threats
Recapitulate tubule geometry and polarity via self-assembly or bioprinting Improved diffusion and culture longevity Scalable and modular for high-throughput screening Use of patient-derived or iPSC-based cells Precise architecture via 3D printing and patterning	Low haploid yield (<1%) and incomplete spermiogenesis Risk of aneuploidy and genetic instability Lack of vasculature and immune components Variability in tubule formation Limited bioactivity in synthetic scaffolds	Development of defined, tunable hydrogels Integration of vasculature and immune cells Combination with perfusion systems Use of gene-editing and reporter tools Platforms for disease modeling and toxicology	Regulatory limitations on scaffold materials and manipulation Faster progress from microfluidics or PSC-based IVG High cost and complexity Proprietary restrictions on bio-inks and printers Potential culture-induced epigenetic defects

Abbreviations: 3D, three-dimensional; iPSC, induced pluripotent stem cell; PSC, pluripotent stem cell; IVG, in vitro gametogenesis.

**Table 3 ijms-26-07383-t003:** SWOT analysis of microfluidic “testis-on-chip” devices and perfused bioreactors.

Strengths	Weaknesses	Opportunities	Threats
Continuous perfusion mimics physiological flow Fine control of gradients and hormone delivery Real-time monitoring and sampling Scalable from chips to bioreactors Multicellular integration (Sertoli, Leydig, etc.) High-throughput and modular systems	No mature human sperm achieved Sensitive to shear stress Material issues (e.g., PDMS absorption) Technical complexity and cost Uneven flow in large bioreactors Short culture durations (<3 weeks)	Vascularized and immune-integrated designs Co-culture with niche cell types Sensor-guided dynamic control GMP-compatible, xeno-free chips Multi-omics profiling of secretions Personalized drug testing and fertility models	PSC-based IVG or organoids may outpace Complex regulatory landscape IP and supply chain limitations Contamination risk with perfusion Unknown long-term safety (genetic/epigenetic) High resource and expertise demands

Abbreviations: PDMS, polydimethylsiloxane; PSC, pluripotent stem cell; IVG, in vitro gametogenesis; GMP, good manufacturing practice.

**Table 4 ijms-26-07383-t004:** SWOT analysis of somatic cell co-culture strategies.

Strengths	Weaknesses	Opportunities	Threats
Native paracrine support (GDNF, SCF, FGF2, VEGF) Flexible formats (2D, ALI, hydrogels, organoids) Human-derived feeders available Improved culture longevity and spermatid yield Potential for autologous, gene-corrected cells	Low efficiency and incomplete maturation Difficult control of cell ratios and spatial structure Poor quality of NOA Sertoli cells Compressed spermiogenesis window; abnormal cells Feeder variability reduces reproducibility	Gene editing or iPSC-derived Sertoli-like cells 3D micro-patterned co-cultures Endocrine signal modulation (e.g., FSH, RA) Addition of supportive niche cells (MSC, endothelium) Standardized GMP-grade feeder lines	Regulatory issues (manipulation and feeder contact) Immunogenicity or tumorigenic risk during ICSI Feeder-free alternatives may outperform IP barriers on feeder technologies Feeder senescence reduces function over time

Abbreviations: ALI, air–liquid interface; FGF2, fibroblast growth factor 2; GDNF, glial cell line-derived neurotrophic factor; GMP, good manufacturing practice; iPSC, induced pluripotent stem cell; MSC, mesenchymal stem cell; NOA, non-obstructive azoospermia; RA, retinoic acid; SCF, stem cell factor.

**Table 5 ijms-26-07383-t005:** SWOT analysis of growth factor-driven differentiation protocols.

Strengths	Weaknesses	Opportunities	Threats
Defined, xeno-free and GMP-compliant media Scalable, modular protocols (2D, bioreactors) Mechanistic clarity for individual factors Compatible with PSC platforms Easy and frequent quality control	Low haploid yield (<20%); rare mature sperm Requires precise timing and dosing Incomplete epigenetic reprogramming Lacks biomechanical support (e.g., BTB, shear) High cost and factor instability	Smart media with biosensor-guided delivery AI-based optimization of factor cocktails Use of epigenetic modulators (e.g., HDAC inhibitors, Vit C) Integration with microfluidics for dynamic control Rapid screening for drugs and toxicants	Organoid or co-culture systems may outperform Regulatory challenges with small molecules IP restrictions on protocols or reagents Variability in protein quality across batches Risk of off-target differentiation or tumorigenicity

Abbreviations: BTB, blood–testis barrier; GMP, good manufacturing practice; HDAC, histone-deacetylase; PSC, pluripotent stem cell; Vit C, vitamin C.

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
