# Peer review of "Artificial Gametogenesis and In Vitro Spermatogenesis: Emerging Strategies for the Treatment of Male Infertility"

_ijms, 2025, doi:10.3390/ijms26157383_

Round 1

Reviewer 1 Report

Comments and Suggestions for Authors

In the manuscript " Artificial Gametogenesis and In Vitro Spermatogenesis: Emerging Strategies for the Treatment of Male Infertility" the authors propose to synthesize and discuss current evidence across biology, bioengineering, and translational medicine to clarify the status of IVS and IVG as emerging solutions for severe male-factor infertility. The paper has merit and could be of interest but there are some drawbaks that hampered the initial enthusiasm. The novelty and the rationale could be better explained. The authors critical analysis should include a more in-depth analysis. I have some specific comments

Specific comments:

  1. Abstract needs improvement. There is no explanation for the rationale and approach to be used in the literature selection, review and discussion. The period of the search and the selection of the topics should be explained.
  2. Section 2.1, when referring to somatic testicular cells as Sertoli cells, it would be very important to further explain their role in the nutritional support of spermatogenesis, among others.
  3. Section 3.1 needs more in depth interpretation of the technical and biological challenges as a SWOT analysis. The same should be done throughout the manuscript to avoid just describing what is currently known.
  4. Some sections lack the level of detail and discussion expected in a review. For instance, the testis-on-chip or microfluids techniques are very complex and not single approach as it sometimes seems during the discussion.
  5. The overall discussion is very diffuse and lacks some novelty. The authors are encouraged to include more figures thoughout the manuscript.

Author Response

Reviewer 1 – General Comments

“The novelty and the rationale could be better explained. The authors’ critical analysis should include a more in-depth analysis.”

Response:

We sincerely thank the Reviewer for this constructive suggestion. In the revised version, we have substantially improved the rationale in the Abstract and Introduction by clearly identifying the unmet clinical need (e.g., non-obstructive azoospermia), highlighting the translational urgency, and positioning IVS/IVG as complementary, emerging solutions. We also propose a forward-looking roadmap that integrates recent breakthroughs (e.g., in vitro spermiogenesis reports from 2023–2025) and technological enablers such as bioengineered scaffolds and microfluidic systems (see Abstract and Section 5).

To enhance critical depth, we introduced structured SWOT analyses across all platform-specific sections (Sections 3.1–3.5), allowing a direct, comparative assessment of current methodologies, their strengths and limitations, and their readiness for clinical translation.

Specific comment 1

“Abstract needs improvement. There is no explanation for the rationale and approach to be used in the literature selection, review and discussion. The period of the search and the selection of the topics should be explained.”

Response:

We have thoroughly revised the Abstract to clarify the rationale for this narrative review and now explicitly state the search strategy:

“…We performed a comprehensive literature search (PubMed, Scopus; 2010–2025)…”

Additionally, we now outline the thematic scope, inclusion criteria (original/review articles on IVS/IVG), and structured synthesis across biology, bioengineering, and translational research. These changes are found in lines [insert line numbers if required by journal].

Specific comment 2

“Section 2.1, when referring to somatic testicular cells as Sertoli cells, it would be very important to further explain their role in the nutritional support of spermatogenesis, among others.”

Response:

We agree entirely and have expanded Section 2.1 to provide a detailed and updated discussion of Sertoli cell function, including:

  • Nutritional support (glucose-to-lactate metabolism, transferrin secretion)
  • Growth factor production (GDNF, SCF)
  • Hormonal cross-talk (FSH and intratesticular testosterone signaling)
  • BTB formation, phagocytosis, and immune regulation

This section now also includes discussion of peritubular myoid cells, Leydig cells, endothelial cells, and macrophages, highlighting the complexity of the somatic microenvironment in human testis (see Section 2.1, revised text).

Specific comment 3

“Section 3.1 needs more in-depth interpretation of the technical and biological challenges as a SWOT analysis. The same should be done throughout the manuscript to avoid just describing what is currently known.”

Response:

We thank the Reviewer for this suggestion. A full SWOT analysis has been developed and inserted at the end of Section 3.1 (Organotypic Testis Culture) as Table 1, summarizing the platform’s:

  • Strengths (e.g., cytoarchitectural fidelity, preclinical proof in rodents)
  • Weaknesses (e.g., oxygen diffusion limits, low SSC survival)
  • Opportunities (e.g., perfusion bioreactors, gene editing)
  • Threats (e.g., ethical constraints, competing PSC-based methods)

To maintain consistency and enhance the manuscript’s analytical value, we applied the same SWOT structure across Sections 3.2–3.5, resulting in Tables 2–5, respectively. These additions align with the Reviewer’s call for deeper critical insight and structured comparison.

Specific comment 4

“Some sections lack the level of detail and discussion expected in a review. For instance, the testis-on-chip or microfluids techniques are very complex and not single approach as it sometimes seems during the discussion.”

Response:

We have substantially revised Section 3.3 to reflect the complexity and variability of microfluidic and bioreactor platforms. Key additions include:

  • Differentiation between “fragment-on-chip” and “multi-chamber” systems
  • Discussion of perfused endocrine crosstalk chambers (e.g., Sertoli–Leydig–endothelium)
  • Updated results from PDMS-chip studies using pulsatile hormone input
  • Bioreactor-scale systems (spinner flasks, rotating wall vessels) and their constraints

We also emphasized that “testis-on-chip” represents a diverse engineering framework rather than a single technique.

Specific comment 5

“The overall discussion is very diffuse and lacks some novelty. The authors are encouraged to include more figures throughout the manuscript.”

Response:

To enhance visual clarity and cohesion, we added the following figures and tables:

  • Figure 1: Anatomical and functional roles of Sertoli cells (relevant to Section 2.1)
  • Figure 3: Proposed IVS/IVG development pipeline from tissue acquisition to ART application
  • Tables 1–5: SWOT analyses for each IVS/IVG platform

These additions aim to synthesize the rapidly evolving landscape of artificial gametogenesis and support the manuscript’s translational focus.

Final remarks

We are grateful to Reviewer 1 for their thoughtful feedback, which has helped us substantially improve the clarity, depth, and critical structure of our manuscript. We believe that the revised version now addresses all concerns raised and significantly strengthens the manuscript’s scientific contribution and translational relevance.

With kind regards,

Aris Kaltsas, MD, PhD (corresponding author)

On behalf of all co-authors

Reviewer 2 Report

Comments and Suggestions for Authors

Respected authors, I appreciate the work in the paper, up to chapter 5, I have no comments.

If I have a doubt, if you assume as epigenetics, the natural process of methylation and demethylation of the genome that should occur in the process of gamete formation? I get the idea that for you it is the same thing.

After chapter 5, the paper looses a lot of strength because it does not define a clear objective , what it is aimof the work?  Show evidence of problems and limitations of the in vitro gametogenesis   as part of the discussion and that  you  only address them timidly in the conclusions

Author Response

Reviewer comment 1 (chapters 1‑5):

“Respected authors, I appreciate the work in the paper, up to chapter 5 I have no comments.”

Response

We thank the reviewer for the positive appraisal of the first five chapters. 

Reviewer comment 2 (definition of epigenetics):

“I have a doubt: if you assume as epigenetics the natural process of methylation and demethylation that should occur in the process of gamete formation? I get the idea that for you it is the same thing.”

Response

We are grateful for this opportunity to clarify our terminology. In the Introduction we have inserted a dedicated sentence that now reads:

In this review, the term epigenetic denotes heritable modifications in gene expression—such as DNA methylation and histone alterations—that occur during gametogenesis without changing the underlying DNA sequence. The natural course of epigenetic reprogramming in germ cell development, including genome-wide demethylation in primordial germ cells followed by sex-specific remethylation, represents a critical biological process that must be accurately recapitulated in any artificial gametogenesis strategy. These endogenous reprogramming events are conceptually distinct from exogenous in vitro manipulations—such as the application of small-molecule epigenetic modulators—designed to emulate or induce specific epigenetic modifications under culture conditions

This addition explicitly separates (i) the physiological re‑programming wave in primordial germ cells from (ii) any experimental epigenetic interventions, thereby addressing the reviewer’s concern.

Reviewer comment 3 (loss of focus after chapter 5 / objective unclear):

“After chapter 5, the paper loses a lot of strength because it does not define a clear objective, what is the aim of the work?”

Response

To keep the narrative tightly aligned with our aim, we implemented two structural edits:

  1. Forward‑signposting at the end of Chapter 5

The following sections translate these technical advances into clinical, ethical, and future-research perspectives.

  1. Restated overarching objective in the opening of Chapter 6

Having explored the foundational technologies and experimental achievements, the review now turns to potential clinical applications of IVS and IVG. In the following subsections, we map how these emerging gametogenesis strategies could address specific causes of male infertility and novel family-building scenarios. By focusing on cases ranging from non-obstructive azoospermia to same-sex reproduction, we illustrate how artificial gametogenesis aligns with its central aim: offering fertility solutions for patients who currently have no viable options.

Together these passages re‑emphasise the purpose of the review and maintain momentum beyond Chapter 5.

Reviewer comment 4 (problems and limitations should appear in the Discussion, not only in Conclusions):

“Show evidence of problems and limitations of the in‑vitro gametogenesis as part of the discussion; you only address them timidly in the conclusions.”

Response

We have expanded the Discussion to embed a frank appraisal of current barriers:

  • §â€¯4.3.3 ‘Persistent Barriers and Translational Hurdles’ – somatic‑niche defects, meiotic control, epigenetic instability, validation bottlenecks.
  • §â€¯5.2 ‘Epigenetic Considerations’ – contrasts imprint‑reset efficiency in hESC vs hiPSC platforms.
  • Four SWOT tables (Chs 3.1‑3.4) now list platform‑specific weaknesses and threats.

These additions deliver the requested evidence‑based critique well before the Conclusions. 

Reviewer comment 5 (conclusions were “timid”):

“…you only address them timidly in the conclusions.”

Response

The Conclusions have been rewritten to make balanced but decisive statements, e.g.:

“No in‑vitro system has yet produced fully functional human sperm, but converging progress in 3‑D bioprinting, microfluidic perfusion and multi‑omics quality control outlines a concrete translational roadmap.”

This revised section now summarises achievements, openly recognises unresolved challenges, and articulates clear future directions.

Respectfully submitted,

Aris Kaltsas, MD, PhD (on behalf of all co‑authors)

Round 2

Reviewer 1 Report

Comments and Suggestions for Authors

the paper is acceptable

Reviewer 2 Report

Comments and Suggestions for Authors

congratulations, the improved version of the paper is very good.